# MrRoPE: Mixed-radix Rotary Position Embedding

**Qingyuan Tian[1], Wenhong Zhu[1], Xiaoran Liu[2], Xiaofeng Wang[1], Rui Wang[1]***

[1]Shanghai Jiao Tong University, [2]Fudan University

{qy.tian, wangrui12}@sjtu.edu.cn

## Abstract

Rotary Position Embedding (RoPE)-extension refers to modifying or generalizing the Rotary Position Embedding scheme to handle longer sequences than those encountered during pre-training. However, current extension strategies are highly diverse and lack a unified theoretical foundation. In this paper, we propose *MrRoPE (Mixed-radix RoPE)*, a generalized encoding formulation based on a radix system conversion perspective, which elegantly unifies various RoPE-extension approaches as distinct radix conversion strategies. Based on this theory, we introduce two training-free extensions, *MrRoPE-Uni* and *MrRoPE-Pro*, which leverage uniform and progressive radix conversion strategies, respectively, to achieve "train short, test long" generalization. Without fine-tuning, MrRoPE-Pro sustains over 85% recall in the 128K-context Needle-in-a-Haystack test and achieves more than double YaRN's accuracy on Infinite-Bench retrieval and dialogue subsets. Theoretical analysis confirms that MrRoPE-Pro effectively raises the upper bound of RoPE's attainable encoding length, which further validates the reliability and utility of our theory and methodology.

## 1 Introduction

Effective understanding of long-text contexts is a cornerstone for advanced NLP tasks (Liu et al., 2023a). For Large Language Models (LLMs), this capability is fundamentally underpinned by the Rotary Position Embedding (RoPE) (Su et al., 2024), which provides a robust foundation for modeling long-range dependencies (Grattafiori et al., 2024; Yang et al., 2025b). In RoPE, the positional information of tokens is encoded through rotation angles, where each embedding dimension is associated with a distinct rotational frequency. Higher dimensions rotate more slowly, which implies that during training, these dimensions often fail to experience a complete cycle. As a result, once the context length exceeds the training window, those high-dimensional features encounter unseen rotation angles, leading to generalization failure (Liu et al., 2023b). To address this issue, early studies proposed continuing training with a larger base frequency, thereby compressing the rotation angles of higher dimensions into the previously observed range and mitigating the occurrence of out-of-domain (OOD) positions (Kazemnejad et al., 2023). However, this strategy requires substantial additional computation (e.g., 57740 GPU hours for Llama2-70B to 32K context window), making larger context extension prohibitively expensive (Touvron et al., 2023; Xiong et al., 2023).

Consequently, recent research has shifted toward training-free approaches to extend the context window of LLMs. Chen et al. (2023) first proposed Position Interpolation (PI), which uniformly scales down the rotation angles across all dimensions to modify RoPE, though the method still requires light fine-tuning on a small amount of data. Alternatively, NTK-aware Interpolation introduced by bloc97 (2023) reduces the distortion in low-dimensional spaces, thereby enabling a substantial expansion of the context window without any extra training. Building upon these insights, Peng et al. (2023) introduced YaRN, an NTK-by-part strategy that applies extrapolation in low dimensions and interpolation in high dimensions, ultimately achieving efficient long-context extension.

Despite the effectiveness of YaRN, we identify a potential limitation in YaRN's approach: *Is linear interpolation used for middle dimensions truly the best choice for maximizing model performance*

---

* Corresponding author

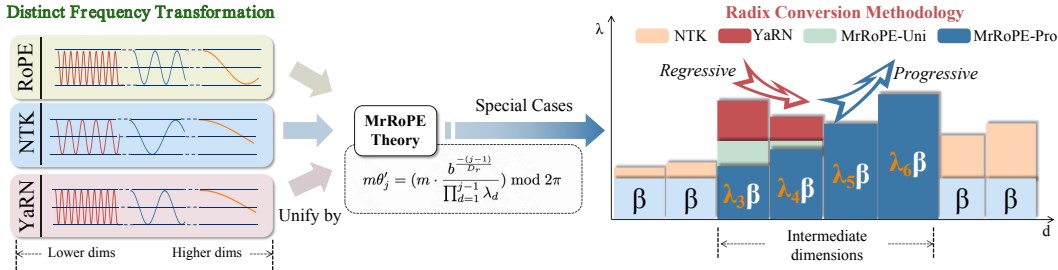

Figure 1: The overall framework of our work. Our key contributions are: (1) a unified theoretical framework for major RoPE-extensions, reflecting them into a specific radix conversion behavior; (2) a progressive radix conversion method MrRoPE-Pro, which outperforms other SoTA methods across various tasks.

*on long sequences?* To answer this question, we first introduce **MrRoPE (Mixed-radix Rotary Position Embedding)**, a unified theoretical framework to generalize existing RoPE-based extension methods (PI, NTK, YaRN ...) and reflect it to a specific radix conversion approach. Guided by this framework, we identify that YaRN's conservative strategy in lower frequency dimensions may excessively disrupt the high-frequency information inherent in the original RoPE encoding. Hence, we further propose two novel RoPE extension schemes, **MrRoPE-Uni** and **MrRoPE-Pro**, which exhibit fundamentally different radix conversion behaviors from YaRN. Their design allows us to systematically compare the efficacy of different radix conversion strategies.

By employing a progressive radix conversion to RoPE, MrRoPE-Pro achieves substantial and consistent gains across both synthetic and real-world long-context evaluations. On the RULER benchmark, it maintains stable accuracy up to 128K tokens, whereas YaRN experiences a sharp degradation beyond 64K. On Infinite-Bench, MrRoPE-Pro not only surpasses YaRN by large margins but also approaches or even exceeds the performance of specialized fine-tuned long-context models—all without additional training. Complementing these empirical findings, our theoretical analysis demonstrates that MrRoPE-Pro significantly enlarges the effective context window predicted by RoPE Bound Theory and stabilizes attention score distributions in intermediate dimensions. Together, these results establish MrRoPE-Pro as a theoretically grounded and empirically robust method for extending the context window of RoPE-based LLMs.

## 2 RoPE AND RADIX THEORY

### 2.1 PRELIMINARY: ROTARY POSITION EMBEDDINGS (ROPE)

Our research is grounded in the RoPE introduced by Su et al. (2024), a position embedding scheme that underpins numerous state-of-the-art LLMs. Consider one specific attention head in a given layer of a Transformer-based model, the set of its hidden neurons is denoted by $D$. Given an input sequence of vectors $\boldsymbol{x_1}, \ldots, \boldsymbol{x_L} \in \mathbb{R}^{|D|}$, it's essential to maintain the relative position information between each $(\boldsymbol{x_m}, \boldsymbol{x_n})$ in the result of the self-attention calculation. To address this issue, RoPE first uses a rotation operation that converts vector $\boldsymbol{x_m}$ into its corresponding query vector $\boldsymbol{q_m}$ and key vector $\boldsymbol{k_m}$ (Vaswani et al., 2017):

$$\{\boldsymbol{q}, \boldsymbol{k}\}_m = f_{\{q,k\}}(\boldsymbol{x_m}, m) = e^{im\boldsymbol{\theta}} \boldsymbol{W}_{\{q,k\}} \boldsymbol{x_m}, \tag{1}$$

where $\boldsymbol{\theta} = \text{diag}(\theta_1, ..., \theta_{|D|/2})$ is a diagonal matrix with $\theta_i = b^{-2(i-1)/|D|}$. $b$ is a predefined value, which is typically set to 10000. In this way, RoPE establishes an injective relation between a token's absolute position and the rotation steps of its embedding vector. Crucially, in the subsequent self-attention computation, the interaction between queries and keys inherently encodes relative positional information as follows:

$$\langle \boldsymbol{q_m}, \boldsymbol{k_n} \rangle = \langle f_q(\boldsymbol{x_m}, m), f_k(\boldsymbol{x_n}, n) \rangle_{\mathbb{R}} = \text{Re}\left(\boldsymbol{x_m^*} \boldsymbol{W_q^*} \boldsymbol{W_k} \boldsymbol{x_n} e^{i\boldsymbol{\theta}(\boldsymbol{m}-\boldsymbol{n})}\right), \tag{2}$$

where $*$ denotes the conjugate transpose operation. For this reason, the attention score depends solely on the relative position $m - n$ rather than the absolute position, which facilitates LLMs in

more effectively capturing the relationships between tokens. In the implementation stage, $e^{im\boldsymbol{\theta}}$ in Eq. 1 can be further expressed as a block diagonal matrix as follows:

$$e^{im\boldsymbol{\theta}} = \mathrm{diag}(\boldsymbol{A}_1, \cdots, \boldsymbol{A}_{\frac{|D|}{2}}); \boldsymbol{A}_j = \begin{pmatrix} \cos m\theta_j & -\sin m\theta_j \\ \sin m\theta_j & \cos m\theta_j \end{pmatrix}. \tag{3}$$

In conclusion, RoPE treats $\boldsymbol{q}$ or $\boldsymbol{k}$ vector as a $\frac{|D|}{2}$- dimensional complex vector, and applies block-wise rotations (i.e., each component will be rotated in its corresponding frequency domain where higher dimension gets slower frequency).

## 2.2 RETHINKING ROPE UNDER THE RADIX THEORY

As we showed above, the key idea of RoPE is to divide the whole $\boldsymbol{q}_m$ (or $\boldsymbol{k}_m$) vector into $D_r$ parts (i.e., $D_r = |D|/2$), and rotate each sub-vector by a corresponding angle $m\theta_j$ (i.e., $j = 1, 2, \cdots, D_r$). Formally, given a non-rotated complex vector $\boldsymbol{W}_{\{q,k\}}\boldsymbol{x_m} \in \mathbb{C}^{D_r}$, its one-dimensional position $m$ is expanded to a $D_r$- dimensional vector $m\boldsymbol{\theta} \in \mathbb{R}^{D_r}$ by RoPE. The rotation angle $m\theta_j$ for $j$-th part can be calculated as follows:

$$m\theta_j = (m \cdot b^{\frac{-(j-1)}{D_r}}) \bmod 2\pi. \tag{4}$$

Most notably, Eq. 4 is strikingly similar to what radix (base) conversion does: representing a number as a sequence of digits. Specifically, given a decimal number $m_{(10)}$, the $j$-th digit (counted from right to left, starting at $j = 1$) of $m_{(\beta)}$ is given by:

$$(m_{(\beta)})_j = \left\lfloor \mathrm{m} \cdot \beta^{-(\mathrm{j}-1)} \right\rfloor \bmod \beta. \tag{5}$$

Then, the relative position $m$ can be calculated as:

$$m = f(m_{(\beta)}) = \sum_{j=1}^{D_r} \beta^{(j-1)}(m_{(\beta)})_j. \tag{6}$$

Eq. 5 indicates that when $\beta = b^{1/D_r}$, it shares a common term $m \cdot \beta^{-(\mathrm{j}-1)}$ with Eq. 4. Remarkably, both the modulo and trigonometric functions contribute to the periodicity. Based on this observation, Su (2023) hypothesizes that if we temporarily disregard the influence of the flooring operation as well as the period size of the modulo operation, RoPE essentially performs a radix conversion from the decimal system to a $b^{1/D_r}$-radix representation.

To prove this hypothesis, we reintroduce the ignored components (the floor function and modulo function) to recover a biased positional estimate $\hat{m}$ from the RoPE as follows:

$$\hat{m} = g(m\boldsymbol{\theta}) = \sum_{j=1}^{D_r} \beta^{(j-1)}(m\theta_j) = \sum_{j=1}^{D_r} b^{\frac{1-j}{D_r}} \left[ (m \cdot b^{\frac{1-j}{D_r}}) \bmod 2\pi \right]. \tag{7}$$

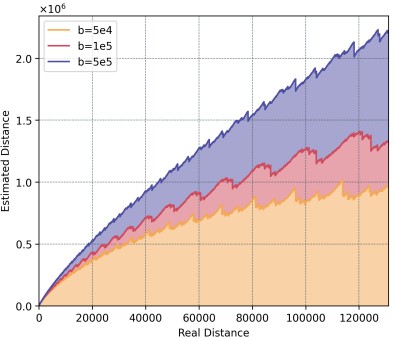

Figure 2: The biased positional estimate $\hat{m}$ of RoPE across different base.

As shown in Figure 2, the biased function recovered from RoPE exhibits a linear trend analogous to that in Eq 6. This linearity is confined to shorter distances for smaller bases, but becomes markedly more pronounced and expands to encompass the full context window as the base increases, suggesting that larger bases possess greater potential for representing positional information over long sequences, which is in accordance with previous findings (Liu et al., 2023b; Men et al., 2024).

**Overall, this section established RoPE as a biased $\beta$-radix encoding by combining theoretical derivation with an empirical linearity analysis.** In the next section, we leverage this radix perspective to unify prior context extension methods.

## 3 MRROPE: A UNIFY FRAMEWORK FOR ROPE EXTENSION

### 3.1 MRROPE THEORY

**Why do we need RoPE extension?** In RoPE, generalization failure of test-length position information stems primarily from the OOD problem of incomplete-cycle dimensions (i.e., dimensions $j$ such that $L/\theta_j < 2\pi$) (Liu et al., 2023b; Ding et al., 2024; Huang et al., 2023). From the radix theory introduced above, this phenomenon directly parallels the behavior of high-digit truncation in a radix encoding system: Consider a $\beta$-radix encoding system, when the input is restricted to the range $[0, L]$, all digit positions starting from the $d$-th and higher never experience a complete carry-over cycle:

$$\left\lfloor L \cdot \beta^{-(j-1)} \right\rfloor \bmod \beta < \beta - 1 \,, \text{ for } j = d, d+1, \cdots \tag{8}$$

A natural solution to scale-up such a unbalanced radix system, is to scale the radix base for digits before dimension $d$: Given a $\beta$-radix, if the $j$-th digit's base is expanded by a factor $\lambda_j$, the representable range of the system is rescaled by a factor of $\prod_{j=1}^{D_r} \lambda_j$, then the $j$-th digit of this expanded system can be expressed as follows:

$$(m_{(\boldsymbol{\lambda}\beta)})_j = \left\lfloor m \cdot \frac{\beta^{-(j-1)}}{\prod_{d=1}^{j-1} \lambda_d} \right\rfloor \bmod (\beta \lambda_j), \tag{9}$$

where m is the target position for embedding. By rescaling the radix base, the extended mixed radix system is able to prolong the rotational period of each dimension, thereby addressing the high-digit OOD challenge as mentioned above.

**Extending RoPE via Mixed Radix Rotary Embedding (MrRoPE).** In RoPE, a similar mixed-radix conversion can also be implemented. By assigning each dimension an independent frequency scaling factor, a mixed-radix RoPE extension is defined as:

$$m\theta_j' = (m \cdot \frac{b^{\frac{-(j-1)}{D_r}}}{\prod_{d=1}^{j-1} \lambda_d}) \bmod 2\pi \,, \{\boldsymbol{q}, \boldsymbol{k}\}_m = e^{im\boldsymbol{\theta}'} \boldsymbol{W}_{\{q,k\}} \boldsymbol{x}_m. \tag{10}$$

The above formulation establishes the **Mixed-Radix RoPE (MrRoPE)** framework: **any RoPE-based extension method conforming to Eq. 10 instantiates a mixed-radix conversion on positional encoding**. The framework's explanatory power stems from its parameterization: the vector $\boldsymbol{\lambda} = \{\lambda_1, \lambda_2, ..., \lambda_{D_r}\}$ defines the radix conversion factors for each dimension. Consequently, MrRoPE posits that the choice of a length extension method is fundamentally equivalent to choosing a specific policy for redistributing positional information across the frequency spectrum via $\boldsymbol{\lambda}$. This provides a unifying lens through which existing methods can be systematically analyzed as constrained instantiations of this general conversion process. We next demonstrate how both NTK-aware scaling and YaRN are naturally recovered under this framework.

**NTK-awre Interpolation under MrRoPE.** Inspired by the Neural Tangle Kernel (NTK) theory (Jacot et al., 2018; Tancik et al., 2020), NTK-aware Interpolation (bloc97, 2023) essentially implements a uniform radix scaling across all dimensions, which can be denoted as:

$$\lambda_j = S^{\frac{1}{D_r - 1}}. \tag{11}$$

By uniformly setting the radix conversion factor across dimensions, NTK avoids the abrupt OOD collapse of the highest dimension—a key to its initial success in training-free context extension.

**NTK-by-parts(YaRN) under MrRoPE.** To further address the OOD challenge, YaRN (Peng et al., 2023) first categorizes all $D_r$ dimensions into high-, mid-, and low-frequency groups based on their rotation progress and applies a differing conversion scheme to them: For both high- and low-frequency dimensions, YaRN develops a non-conversion strategy, setting $\lambda_j = 1$ to better preserve corresponding position information. Besides, for mid-frequency dimensions (i.e., $d_l \leq j < d_h$), YaRN uses a linear interpolation strategy such that the expansion factor of each radix satisfies:

$$\prod_{d=d_l}^{j-1} \lambda_d = r_j S, \tag{12}$$

where $r_j \in (0, 1)$ is a linear scale factor determined by YaRN. More details about YaRN are presented in Appendix A.2.

## 3.2 MRROPE METHODOLOGY

Based on the empirical evidence from YaRN, we conjecture that an effective radix conversion algorithm should adhere to the following principle: *lower dimensions extrapolate, higher dimensions interpolate, while intermediate dimensions achieve the desired extension factor $S$.*

Although YaRN's formulation does not explicitly define a radix conversion strategy for its intermediate dimensions, we find it implicitly fulfills the above principle through a regressive scaling conversion (i.e., $\lambda_j > \lambda_{j+1}$), as proved in Appendix A.2.1. This observation raises a fundamental question: *is this the optimal conversion strategies for intermediate dimensions?*

Within the MrRoPE framework, we can systematically discuss this question by explicitly designing the $\boldsymbol{\lambda}$ vector. We design two distinct strategies: Uniform Conversion, which applies a constant radix factor (i.e., $\lambda_j = \lambda_{j+1}$); and Progressive Conversion, which features a monotonically increasing scaling factor (i.e., $\lambda_j < \lambda_{j+1}$).

Based on these strategies, we are able to implement two additional radix transformation methods distinct from YaRN: ***MrRoPE-Uni*** and ***MrRoPE-Pro***, and find a better conversion for intermediate dimensions through comparison.

### 3.2.1 MRROPE-UNI

NTK pioneered the concept of uniform radix conversion for context extension. However, its failure to account for spectral distinctions leads to high-frequency distortion and persistent OOD errors. Accordingly, MrRoPE-Uni adopts a segmentation-based NTK approach in which a constant scaling factor is applied for uniform radix conversion.

Let $\lambda_j = c$ for all $j \in [d_l, d_h)$, where $c$ is a constant. To achieve the total scale factor $S$, the radix expansion factor for intermediate dimension can be calculated as:

$$\lambda_j = S^{\frac{1}{d_h - d_l}}.\tag{13}$$

In summary, MrRoPE-Uni applies a uniform scaling to all middle dimensions, independent of their original frequencies, effectively expanding the positional encoding range by a factor of $S$.

### 3.2.2 MRROPE-PRO

Motivated by high-frequency extrapolation, MrRoPE-Pro employs a progressive radix conversion with dimension-wise scaling. In this scheme, lower (high-frequency) dimensions undergo smaller radix expansions, while higher (low-frequency) dimensions experience proportionally larger expansions. Hence, MrRoPE-Pro better preserves the fine-grained structure of high-frequency dimensions while enabling effective extension of the representable positional range, thus providing a more faithful extrapolation in the intermediate regions.

Let $\lambda_j = S^{\epsilon_j}$. To obtain a progressive sequence, we assume that the sequence $\epsilon$ follows an arithmetic progression. Let $\epsilon_{d_l-1} = 0$ and $\epsilon_j - \epsilon_{j-1} = c$, where $c$ is a constant for all $j \in [d_l, d_h)$, it is straightforward to obtain that $\epsilon_j = (j - d_l + 1)c$ for middle-dimensions. According to Eq. 12, which implies the constraint $\sum \epsilon_j = 1$, the following expression for $\epsilon_j$ in the middle dimensions can be derived:

$$\epsilon_j = \frac{2(1 + j - d_l)}{(1 + d_h - d_l)(d_h - d_l)}, \lambda_j = S^{\epsilon_j}.\tag{14}$$

In summary, MrRoPE-Pro applies a progressive scaling to all middle dimensions, resulting in a slow-to-steep radix conversion that better preserves the positional information encoded in the lower (high-frequency) dimensions.

### 3.2.3 GENERAL FORMULATION

**Overall Formulation of MrRoPE** can be formulized as:

$$m\theta'_j = (m \cdot \frac{b^{\frac{-(j-1)}{D_r}}}{\prod_{d=1}^{j-1} \lambda_d}) \bmod 2\pi\tag{15}$$

where $D_r = |D|/2$. For the radix expansion factor $\lambda_d$ on $d$-th dimension, we define:

- **For low- and high- dimensions**, the scale factor $\lambda_d = 1$ (i.e., $d \in [1, d_l) \cup [d_h, D_r]$).

- **For middle-dimensions**, the scale factor $\lambda_i$ values are determined by the method used as follows:

$$\lambda_d = \begin{cases} S^{\frac{1}{d_h - d_l}}, & \text{if } \textit{MrRoPE-Uni} \\ S^{\frac{2(1+d-d_l)}{(1+d_h-d_l)(d_h-d_l)}}, & \text{if } \textit{MrRoPE-Pro} \end{cases} \quad (d_l \le d < d_h). \tag{16}$$

Additional tricks for implementations are consistent with YaRN, such as the value of $d_l$ and $d_h$, which are provided in Appendix B. Evidently, the fundamental distinction between our method and YaRN resides exclusively in the extrapolation strategy employed for the intermediate dimensions. Figure 3 shows the cumulative scaling factor $s_d$ for each dimension across various context window extension methods, where $s_d = \prod_{j=1}^{d-1} \lambda_j$. Notably, MrRoPE-Uni and MrRoPE-Pro adopt distinct strategies in the middle dimensions compared to YaRN. In the following experiments, we treat YaRN as a special case of MrRoPE-Regressive, using it as a baseline for performance comparison.

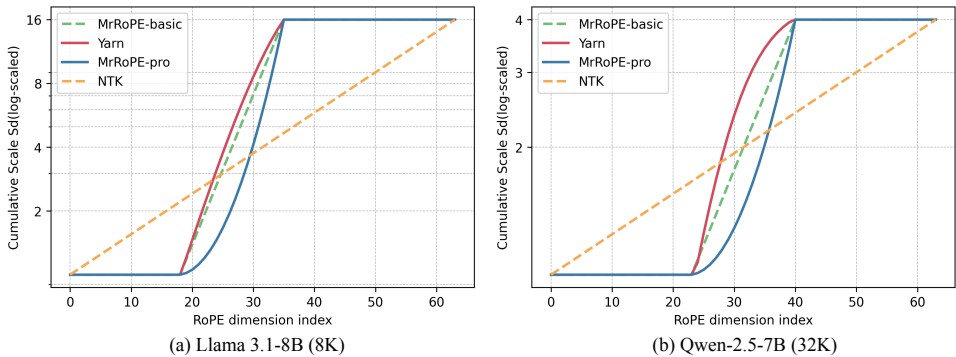

(a) Llama 3.1-8B (8K)          (b) Qwen-2.5-7B (32K)

Figure 3: The cumulative scaling factor $s_d$ of different RoPE extension methods across varying dimension index.(Scale-up to 16x and 4x)

## 4 EXPERIMENTS

### 4.1 SETTINGS

**Baselines**. We compare MrRoPE with SOTA test-time RoPE-extension methods, including YaRN, NTK. All experiments were conducted in an inference setting (no fine-tuning), and thus, no comparisons were conducted with other context extension methods that require training (Shang et al., 2025; Wang et al., 2024; Hua et al., 2024).

**Base Models and Tasks**. We evaluate MrRoPE on the following RoPE-based LLMs: LLaMA2-7B (Touvron et al., 2023), LLaMA3-8B (Grattafiori et al., 2024), and Qwen2.5-3B (Yang et al., 2025a). Our evaluation mainly addresses three aspects: (1) perplexity curves on a pre-training test set to assess performance under extended context; (2) long-context stress tasks, including RULER (Hsieh et al., 2024) and Needle-In-a-Haystack (Kamradt, 2023), to evaluate the handling of long-range dependencies; and (3) real-world benchmarks, Infinite-Bench (Zhang et al., 2024) and Longbench-v2 (Bai et al., 2024) to examine model performance in practical long-context scenarios.

**Theoretical Analysis**. Besides the regular tests introduced above, we also provide theoretical evidence supporting the reliability of MrRoPE. Inspired by prior work (Liu et al., 2023b; Barbero et al., 2024), we first investigate the attention scores, with a particular focus on their performance improvement in the middle dimensions. Moreover, based on the theory of Men et al. (2024), we employ the cosine similarity metric to assess the enhancement in the theoretical maximum encoding length achieved by MrRoPE.

### 4.2 LONG SEQUENCE LANGUAGE MODELING

Long-sequence modeling serves as the most direct and fundamental evaluation of positional encoding methods. To assess this ability, we compute perplexity scores on ten randomly sampled

sequences exceeding 128K tokens from the Proofpile dataset (Azerbayev et al., 2022), using float32 precision. Table 1 reports results across different base models and extension strategies; due to the space limitation, perplexity results of LLaMA2-7B are provided in Appendix C.1.

Table 1: Perplexity scores on proofpile dataset across different models. The best and second-best results are boldfaced and underlined, respectively.

| (a) LLaMA3-8B-Instruct | | | | | | | | (b) Qwen2.5-3B-Instruct | | | | | | | |
|---|---|---|---|---|---|---|---|---|---|---|---|---|---|---|---|
| Context Window | Scale (S) | Extension Method | Evaluation Context Length | | | | | Context Window | Scale (S) | Extension Method | Evaluation Context Length | | | | |
| | | | 8K | 16K | 32K | 64K | 128K | | | | 8K | 16K | 32K | 64K | 128K |
| 8K | 16 | NTK | 3.67 | 3.72 | >5 | >10 | >10 | 32K | 4 | NTK | 3.548 | 3.778 | 7.782 | >10 | >10 |
| | | YaRN | 3.68 | 3.08 | 2.75 | 2.49 | _2.38_ | | | YaRN | 3.449 | 2.894 | 2.589 | 2.333 | _2.222_ |
| | | MrRoPE-Uni | 3.66 | 3.06 | _2.74_ | _2.47_ | 2.41 | | | MrRoPE-Uni | _3.440_ | _2.885_ | _2.581_ | _2.329_ | 2.224 |
| | | MrRoPE-Pro | **3.63** | **3.03** | **2.71** | **2.45** | **2.34** | | | MrRoPE-Pro | **3.433** | **2.881** | 3.579 | **2.328** | **2.221** |

**YaRN does not consistently achieve superior performance across models and context lengths**. On the Proofpile test set, YaRN yields markedly higher perplexity in the initial half of extended contexts, where both MrRoPE-Uni and MrRoPE-Pro consistently outperform it across model families. Moreover, as the context approaches the end of the extended window, YaRN begins to surpass MrRoPE-Uni, yet remains inferior to MrRoPE-Pro. For instance, at a 128K context length, LLaMA3-8B records a perplexity of 2.38, surpassing MrRoPE-Uni (2.41) but remaining substantially weaker than MrRoPE-Pro (2.34). Since similar trends are also observed in both Qwen2.5 and LLaMA2, we attribute this behavior to the regressive radix conversion strategy employed by YaRN: radix conversion in lower-dimensional spaces disrupts local positional information, degrading performance in shorter contexts while partially preserving longer-range coherence. These findings indicate that although YaRN alleviates certain challenges in long-sequence modeling, it incurs trade-offs that diminish its overall robustness across the full context window.

**MrRoPE-Pro demonstrates superior and consistent performance across the entire extended context window**. On both LLaMA3-8B and Qwen2.5-3B, MrRoPE-Pro steadily achieves the lowest perplexity across all evaluated lengths, ranging from 8K to 128K. The merit is particularly evident in the shorter-range context: for instance, on LLaMA2-7B at 4K, MrRoPE-Pro attains a perplexity of 5.72, outperforming YaRN (6.02) and MrRoPE-Uni (5.84). This superior performance can be attributed to the progressive extension strategy, which avoids out-of-distribution positional embeddings in higher dimensions while retaining high-frequency details in the original RoPE structure. By mitigating distortions in both local and global positional information, MrRoPE-Pro provides a more balanced and effective approach to context window extension.

Considering the weaker long-context performance of MrRoPE-Uni than the progressive one, we mainly focus on MrRoPE-Pro and YaRN in the following experiments.

### 4.3 LONG CONTEXT BENCHMARKS

To further evaluate real-world long-context reasoning capabilities, we assess model performance on standardized benchmarks. We mainly focus on two key capabilities: (1) *the ability to retrieve relevant information*, and (2) *the ability to utilize such information for real-world tasks*.

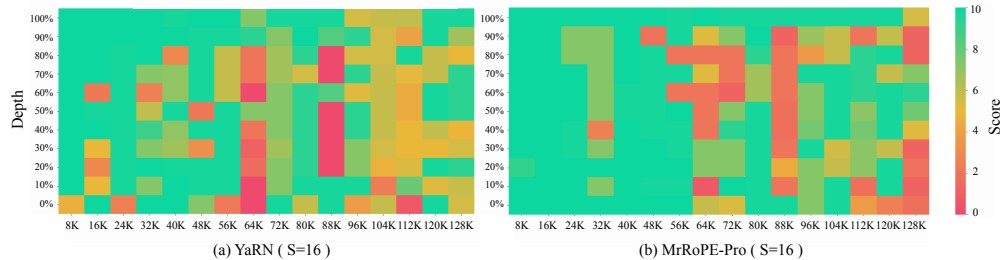

Figure 4: In the Needle-IN-A-Haystack test, MrRoPE-Pro (right) effectively extends LLaMA3-8B's context window to nearly 96K, which is much longer than the performance of YaRN (left).

**MrRoPE-Pro enables robust context window extension in long-context stress task**. To assess the retrieval ability mentioned above, we apply the NIAH test (Kamradt, 2023) with varying context lengths and insertion depths. We compare MrRoPE-Pro, our best-performing method, against YaRN on LLaMA3-8B, measuring recall with ROUGE-1 (Lin, 2004). Figure 4 shows that MrRoPE-Pro extends the effective context window to at least 96K tokens, exhibiting markedly more stable scaling. Within shorter contexts (8K–56K), it consistently achieves higher scores, approaching near-perfect accuracy at insertion depths of 80–100%. Beyond 96K, the performance gap widens: MrRoPE-Pro maintains clear superiority across the 30–70% range, demonstrating robustness in mid-context retrieval under ultra-long settings. Even at 120K tokens (15× the pre-training length), it sustains over 85% recall at most depths, establishing a strong upper bound for practical long-context applications.

We further evaluate MrRoPE on the RULER benchmark, with results presented in Table 2. Notably, MrRoPE demonstrates substantially superior performance across the extended context window. This advantage is particularly pronounced from 64K to 128K tokens, where YaRN's performance sharply declines from 89.5 to 79.9, while MrRoPE maintains a relatively stable level of performance.

Table 2: Retrieval scores on RULER benchmark across all 13 subtasks.

| (a) LLaMA3-8B-Instruct | | | | | | | (b) Qwen2.5-3B-Instruct | | | | | | |
|---|---|---|---|---|---|---|---|---|---|---|---|---|---|
| Scale Setting | Extension Method | Evaluation Context Length | | | | | Scale Setting | Extension Method | Evaluation Context Length | | | | |
| | | 8K | 16K | 32K | 64K | 128K | | | 8K | 16K | 32K | 64K | 128K |
| 8K →128K | YaRN | 95.5 | 92.1 | 92.7 | 89.5 | 79.9 | 32K →128K | YaRN | 78.1 | 77.7 | 75.6 | 63.2 | 50.1 |
| | MrRoPE-Pro | **96.2** | **94.2** | **94.3** | **91.3** | **86.6** | | MrRoPE-Pro | **82.3** | **82.9** | **78.5** | **70.4** | **53.2** |

**MrRoPE-Pro effectively utilizes the inherent extrapolation capabilities of the base model**. To assess MrRoPE-Pro's ability to integrate retrieved context, we first evaluate it on the Infinite-Bench with sub-tasks under 128K tokens, sampling 100 examples per subset. On simpler tasks, MrRoPE-Pro performs competitively with closed-source models, achieving 100% on Passkey Retrieval (matching GPT-4) and 89% on Number Retrieval. In more complex tasks, such as KV Retrieval and QA Dialogue, it significantly outperforms YaRN (**27**% vs. 9% and **22**% vs. 10%, respectively) and even surpasses specialized long-context models like Yi-34B-200K (Young et al., 2024) and Kimi-Chat (Team et al., 2025) on several subsets. These results show that MrRoPE-Pro not only outperforms open-ended methods like YaRN but also narrows the gap with fine-tuned long-context models—without requiring additional training. In addition, we also report the performance of MrRoPE-Pro and YaRN on LongBench-V2, another real-world long-context benchmark with more complex test cases, where MrRoPE-Pro also outperforms YaRN, the results are shown in Appendix C.2.

Table 3: Long context performance comparison on Infinite-Bench. In each subset, we randomly choose 100 samples with lengths ranging from 100K to 128K. MrRoPE-Pro outperforms YaRN under the same settings while approaching GPT-4 in some tasks (e.g., QA Dialogue, Math Find).

| Models | Extensioin Method | Retrieve KV | Retrieve PassKey | Retrieve Number | Code Debug | Math Find | QA Dialogue | Avg. |
|---|---|---|---|---|---|---|---|---|
| GPT-4 | - | **89%** | **100%** | **100%** | **35%** | **60%** | **26%** | **68.3%** |
| Yi-34B-200K | - | 2% | 100% | 100% | 12% | 23% | 2% | 39.8% |
| Kimi-Chat | - | 54% | 98% | 95% | 18% | 13% | 12% | 48.3% |
| LLaMA3-8B | YaRN | 9% | 100% | 85% | 3% | 57% | 10% | 44% |
| LLaMA3-8B | MrRoPE-Pro | **27%** | **100%** | **89%** | **3%** | **58%** | **22%** | **49.8%** |

## 4.4 THEORETICAL ANALYSIS

This section establishes the theoretical basis for MrRoPE-Pro's extrapolation capabilities by analyzing its merits for context window extension. The analysis proceeds in two parts: first, we examine the mechanism responsible for its superior extrapolation limit; second, we demonstrate how this design enlarges the effective context window in the self-attention perspective.

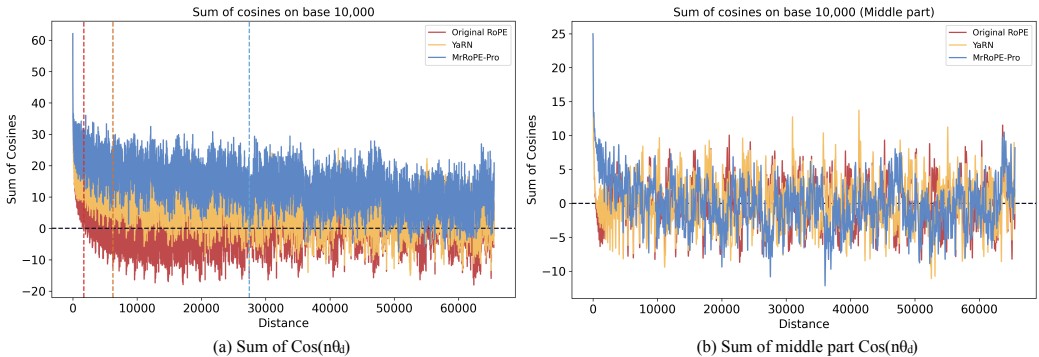

(a) Sum of Cos(nθ_d)

(b) Sum of middle part Cos(nθ_d)

Figure 5: The cosine sum of the rotation angles in each dimension, measuring the ability to give more attention to similar tokens than a random one. The base value and original context length are consistent with the settings of LLaMA2-7B.

**MrRoPE-Pro significantly extends the theoretical context window upper-bound of the base model**. The RoPE Bound Theory establishes a method for analyzing the potential context length of LLMs by relating it to the rotational base $b$. This theory demonstrates that the upper bound of the effective context window is defined by the root of the function $B_\theta(m) = \sum_{j=1}^{D_r} cos(m\theta_j)$. shows the evolution of $B(m)$ with relative distance for various extension methods, with the vertical dashed line indicating this root. Under a base of 10,000, MrRoPE-Pro extends the theoretical context window upper bound from 1K to 28K—nearly five times the 6K bound achieved by YaRN. As the key difference between methods lies in their middle-dimensional strategies, Figure 5b isolates the contribution of these dimensions to $B_\theta(m)$. Evidently, MrRoPE exhibits significantly smaller oscillations and a more stable trend than alternative methods.

**MrRoPE-Pro operates by refining intermediate-dimensional features to stabilize attention score distributions**. In large language models, attention score variation is a key metric for evaluating RoPE-based extension methods. Therefore, to compare MrRoPE-Pro with YaRN, we analyze changes in attention scores across middle dimensions with respect to relative distance. As shown in Figure 6, MrRoPE-Pro extends the periodicity of the original RoPE frequency signals in middle dimensions, maintaining monotonic attention variation over a broader range. This improves the consistency of positional information encoding. Moreover, MrRoPE-Pro produces attention distributions that better match pre-training patterns compared to YaRN, enhancing training-free extrapolation.

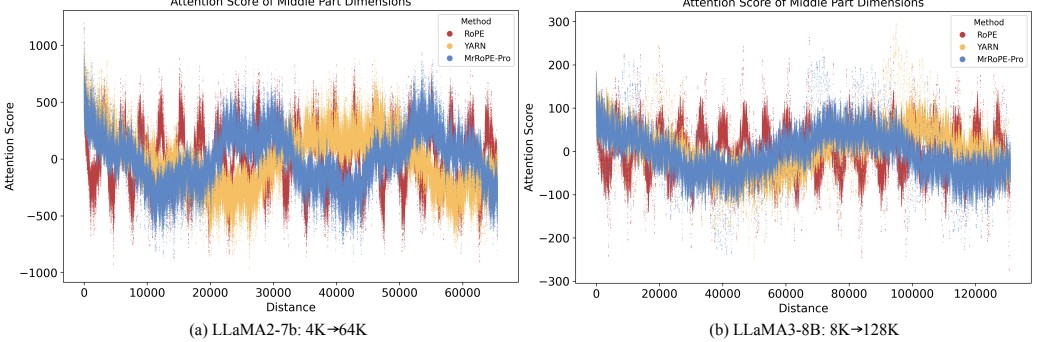

(a) LLaMA2-7b: 4K→64K

(b) LLaMA3-8B: 8K→128K

Figure 6: Middle-partial attention score on extended context windows. For each relative position, we randomly selected 50 token pairs' corresponding attention score calculated by the middle dimensions.

## 5 CONCLUSION

We present a unified theory, MrRoPE, linking major RoPE-extension methods to radix conversion, where existing methods emerge as special cases. Based on this, we propose MrRoPE-Pro, a training-free method that effectively mitigates the high-dimensional OOD problem by progressively scaling

the radix base. Empirical results on real-world benchmarks show that MrRoPE-Pro nearly doubles the practical context window and surpasses strong baselines, including closed-source models. Theoretically, our improvement stabilizes the attention scores in intermediate dimensions and maximally restores high-frequency information, thereby pushing the upper bound of the effective context window to its maximum.

## ETHICS STATEMENT

This work presents a context window extension theory and methodology for Large Language Models via positional encoding. Our research is based on publicly available, pre-trained base models (e.g., LLaMA series) and standard, open-source benchmarks. All data used for evaluation are in the public domain and contain no private or sensitive information. Our work does not introduce new data collection practices or target specific, sensitive applications.

## REPRODUCIBILITY STATEMENT

To facilitate reproducibility, the source code for MrRoPE-Uni/Pro, along with the baseline method YaRN and NTK, has been anonymized and submitted as supplementary materials with this paper. Detailed descriptions of the experimental configuration and all hyperparameters are provided in the Appendix. Our experiments utilize standard, open-source benchmarks and their corresponding official test scripts to ensure all results are easily verifiable.

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

## A RELATED WORKS

### A.1 PI

In the study of RoPE-extension, earlier works seek to mitigate incomplete-cycle effects by re-scaling or normalizing position indices before applying sinusoidal transformations. From the radix perspective, these methods can be understood as normalizing positions to the original trained range so that higher-order digits are less likely to terminate prematurely. A representative example is the Position Interpolation (PI) method (Chen et al., 2023), whose attention score can be calculated as:

$$m' = \frac{m}{S} , \{\boldsymbol{q}, \boldsymbol{k}\}_m = f_{\{q,k\}}(\boldsymbol{x}_m, m') = e^{im'\boldsymbol{\theta}} \boldsymbol{W}_{\{q,k\}} \boldsymbol{x}_m \tag{17}$$

where $S = L_{test}/L_{train}$. Remarkably, because PI method can not be expressed in the form of Eq. 16, we cannot interpret PI as any form of base conversion behavior. This also provides evidence that PI constitutes a normalization of positional values, allowing its underlying logic to still be explained within the framework of MrRoPE. Nevertheless, PI uniformly compresses the digit representation of positions, causing collisions in lower-order digits and thus disproportionately distorting fine-grained positional information.

### A.2 DETAILS OF YARN

YaRN can be considered as a further version of NTK, which applies a segmented NTK-aware Interpolation into different parts. More specifically, YaRN define $\lambda_d$ as the wavelength of the RoPE embedding at $d$-th hidden dimension:

$$T_d = \frac{2\pi}{\theta_d} = 2\pi b^{\frac{2d}{|D|}} \tag{18}$$

It's easy to classify all dimensions into two categories: 1) $L_{train} > T_d$ and 2) $L_{train} \leq T_d$. Liu et al. (2023b) observed that in the first type of dimensions, each position goes through one or more full rotation cycles during training, so the model sees all positional patterns and learns them well. These dimensions remain stable at inference and do not cause out-of-distribution issues. In contrast, the second type of dimensions lacks sufficient training exposure, making them the main cause of extrapolation errors and abnormal attention scores. Therefore, YaRN fully accounts for the differences in wavelengths across dimensions and applies different extrapolation strategies accordingly: it uses

direct extrapolation for well-trained dimensions, periodic interpolation (PI) for those trained for less than one full cycle, and linear approximation for the remaining dimensions as follows:

$$
g\left(m\right) = (1 - \gamma(r_d))\frac{m}{s} + \gamma(r_d)m
$$
$$
h\left(\theta_d\right) = \theta_d
$$

(19)

where $r_d = L/T_d$ and

$$
\gamma(r_d) = \begin{cases} 0, & \text{if } r < \alpha \\ 1, & \text{if } r > \beta \\ \frac{r-\alpha}{\beta-\alpha}, & \text{otherwise} \end{cases}
$$

(20)

In real coordinates, good values for $\alpha$ and $\beta$ are $\alpha = 1$ and $\beta = 32$. In addition, YaRN introduces a further scale factor $t$ to modify the computation of attention weights into

$$
\text{softmax}\left(\frac{\mathbf{q}_m^T\mathbf{k}_n}{t\sqrt{|D|}}\right)
$$

where $t$ can be calculated as $\sqrt{\frac{1}{t}} = 0.1\ln(s) + 1$.

### A.2.1 YaRN is a regressive radix conversion

Considered a middle-dimension $j \in [d_l, d_h)$, based on YaRN's scaling equation, we have

$$
m\theta'_j = (m \cdot b^{\frac{-(j-1)}{D_r}} \cdot \frac{\beta + (s-1)r_j - s\alpha}{s(\beta-\alpha)}) \bmod 2\pi \,, \{\boldsymbol{q},\boldsymbol{k}\}_m = e^{im\boldsymbol{\theta}'}\boldsymbol{W}_{\{q,k\}}\boldsymbol{x}_m.
$$

(21)

According to the MrRoPE theory, the radix expansion factor $\lambda_j$ satisfies

$$
\prod_{d=1}^{j-1}\lambda_d = s \cdot \frac{\beta - \alpha}{\beta + (s-1)r_j - s\alpha}.
$$

(22)

Hence, $\lambda_j$ satisfies

$$
\begin{aligned}
\lambda_j &= \frac{\prod_{d=1}^{j}\lambda_d}{\prod_{d=1}^{j-1}\lambda_d} = \frac{\beta + (s-1)r_j - s\alpha}{\beta + (s-1)r_{j+1} - s\alpha} \\
&= \frac{c + (s-1)r_j}{c + (s-1)r_{j+1}},
\end{aligned}
$$

(23)

where $c = \beta - s\alpha$. Then divide $\lambda_j$ by $\lambda_{j-1}$,

$$
\begin{aligned}
\frac{\lambda_j}{\lambda_{j-1}} &= \frac{(c + (s-1)r_j)^2}{(c + (s-1)r_{j+1})(c + (s-1)r_{j-1})} \\
&= \frac{c^2 + (s-1)c \cdot 2r_j + (s-1)^2r_j^2}{c^2 + (s-1)c \cdot (r_{i+1} + r_{j-1}) + (s-1)^2r_{j+1}r_{j-1}}.
\end{aligned}
$$

(24)

Notably, according to Eq. 18, we have $r_j^2 = r_{j+1}r_{j-1}$. Therefore, the Eq. 25 can be further simplified as follows:

$$
\begin{aligned}
\frac{\lambda_j}{\lambda_{j-1}} &= \frac{c^2 + (s-1)c \cdot 2r_j + (s-1)^2r_j^2}{c^2 + (s-1)c \cdot (r_{j+1} + r_{j-1}) + (s-1)^2r_{j+1}r_{j-1}} \\
&= \frac{c^2 + (s-1)^2r_j^2 + (s-1)c \cdot 2r_j}{c^2 + (s-1)^2r_j^2 + (s-1)c \cdot (r_{j+1} + r_{j-1})} \\
&= \frac{c^2/r_j + (s-1)^2r_j + (s-1)c \cdot 2}{c^2/r_j + (s-1)^2r_j + (s-1)c \cdot (b^{\frac{1}{D_r}} + b^{\frac{-1}{D_r}})} \\
&\leq 1.
\end{aligned}
$$

(25)

In practice, RoPE's original base $b$ is not allowed to be setted as 1. Therefore, for all rope-theta settings where $b > 1$ or $0 < b < 1$, YaRN's scale factors always satisfy $\lambda_{j-1} > \lambda_j$, which characterizes YaRN as a regressive conversion.

# B  DETAILS OF MRROPE METHODOLOGY

In MrRoPE-Pro(Uni) implementation, the calculation of $d_l$ and $d_h$ is the same as that of YaRN's. More specifically, $d_l$ and $d_h$ represent the boundaries that distinguish different types of dimensions, and they can be calculated as follows:

$$\begin{aligned} d_l &= \max\left\{i \mid L_{train} \cdot \theta_i > \beta \cdot 2\pi\right\} \\ d_h &= \min\left\{i \mid L_{train} \cdot \theta_i < \alpha \cdot 2\pi\right\} \end{aligned}$$
(26)

where $\theta_i = b^{-2(i-1)/|D|}$, $\alpha$ and $\beta$ are hyper-parameters. In real coordinates, good values for them are $\alpha = 1$ and $\beta = 32$ (Details are presented in the following section). Another trick used in YaRN, the rescaleing factor $t$ is also encouraged to unlock the full potential of MrRoPE.

## B.1  HYPERPARAMETERS GUIDELINE

sAs we introduced above, changing $(\alpha, \beta)$ is essentially equivalent to selecting a different dimension partition, i.e., modifying $(d_l, d_h)$, and there is a one-to-one correspondence between the two pairs. Therefore, in the following experiments, we treat $d_l$ and $d_h$ as the variables and analyze how the positional encoding performance changes accordingly.

First, the purpose of defining $d_h$ is to isolate dimensions whose rotations do not complete a full cycle during training, which YaRN identifies as the main source of OOD failures. From this view, $d_h$ should be highly sensitive, and increasing it should sharply degrade long-context extrapolation. To test this, we evaluated how PPL changes with $d_h$ on Llama3-8B-Instruct, using $d_h = 35$ (i.e., $\beta = 1$) as the reference point.

As shown in Figure 7a, we observe that the effect of position embedding is not such highly sensitive to $d_h$; the model maintains low PPL within the range $[34, 39]$, and MrRoPE-Pro consistently outperforms YaRN. A similar trend appears when varying $d_l$, as shown in Fig 7b.

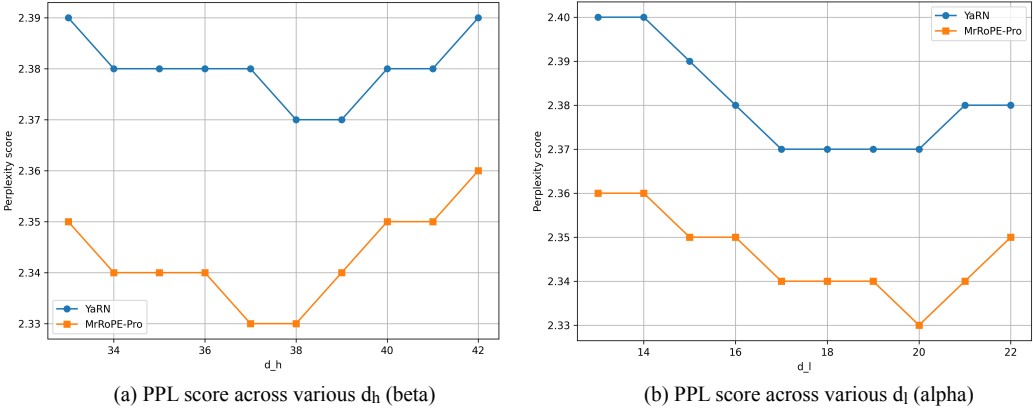

(a) PPL score across various $d_h$ (beta)  (b) PPL score across various $d_l$ (alpha)

Figure 7: LLama3-8B-Instruct PPL score tested on 128K context window across different $\alpha/\beta$ settings.

In addition, we also conducted experiments on Qwen2.5-3B-Instruct to verify whether the same hyperparameters can be applied to models with different architectures. The results are shown in Fig 8. We find that for Qwen2.5-3B-Instruct, the best values of $(d_l, d_h)$ are $(23, 40)$, corresponding to $\alpha = 32$ and $\beta = 1$.

In summary, these results show that different models may prefer different optimal hyperparameter settings, but MrRoPE-Pro is relatively insensitive to these choices. In particular, $\alpha = 32$ and

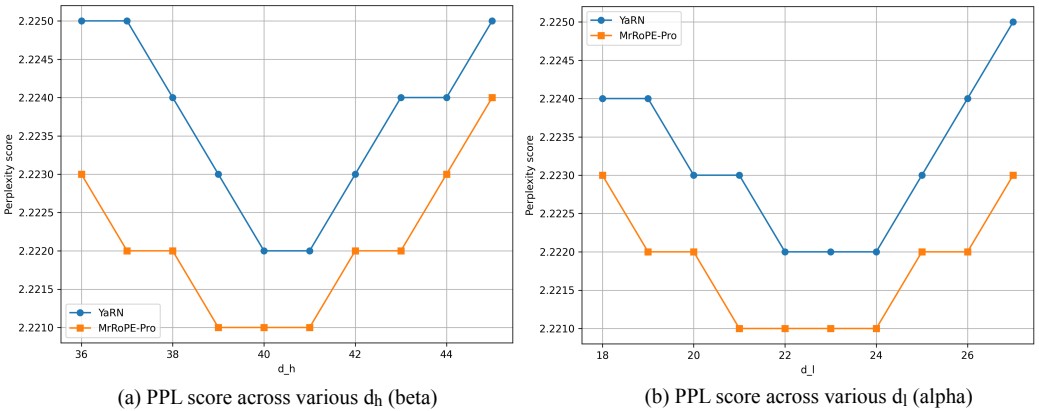

(a) PPL score across various d$_h$ (beta)  (b) PPL score across various d$_l$ (alpha)

Figure 8: Qwen2.5-3B-Instruct PPL score tested on 128K context window across different $\alpha/\beta$ settings.

$\beta = 1$ consistently yield near-optimal performance across models and can serve as a strong default configuration.

## C  ADDITIONAL EXPERIMENTS RESULTS

### C.1  PPL RESULTS OF LLAMA2-7B-CHAT-HF

LLaMA2-7b-chat-hf's perplexity score is as follows:

Table 4: Perplexity scores of LLaMA2-7B-chat-hf on proofpile dataset. The best and second best results are boldfaced and underlined respectively.

| **(a) LLaMA2-7B-Instruct** | | | | | | | |
|---|---|---|---|---|---|---|---|
| Context Window | Scale (S) | Extension Method | \multicolumn Evaluation Context Length | | | | |
| | | | 4K | 8K | 16K | 32K | 64K |
| 4K | 16 | NTK | 6.73 | 6.76 | 6.97 | 7.11 | >10 |
| | | YaRN | 6.02 | 5.59 | 4.64 | 4.07 | 4.03 |
| | | MrRoPE-Uni | 5.84 | 5.42 | **4.52** | **4.01** | 4.15 |
| | | MrRoPE-Pro | **5.72** | **5.39** | 4.57 | 4.11 | **3.88** |

### C.2  LONGBENCH-V2 RESULTS

To test the RoPE-extension method on LongBench-V2, we first limit the length of the test sample to less than 128K to match the extended window length. The performance results are shown in Table 5.

## D  USE OF LLMS

To better present our work, the authors used ChatGPT solely for grammatical checking of the manuscript. The initial draft of the manuscript and all code were created without AI assistance. Its use was strictly limited to identifying and correcting grammatical errors in pre-written text. The prompts were exclusively in the form of: "As a senior grammar editor, you are tasked with conducting a thorough grammatical and syntactical review of the manuscript, ensuring accuracy in all aspects of grammar, including tenses, voices, subject-verb agreement, and clause structures. Pay special attention to the correct use of complex sentences and passive voice common in academic

Table 5: Resultes of LLaMA3-8B-Instruct and Qwen2.5-3B-Instruct on LongBenchV2 dataset. SD: Single-Document QA, LD: Long-dialogue History Understanding, MD: Multi-Document QA, LICL: Long In-context Learning, LSD: Long Structured Data Understanding, CU: Code Repository Understanding.

**(a) LLaMA2-7B-Instruct**

| Context Window | Scale (S) | Extension Method | SD | LD | MD | LICL | LSD | CU |
|---|---|---|---|---|---|---|---|---|
| 4K | 16 | YaRN | 16.7 | 12.8 | 17.8 | 26.8 | 30.0 | 6.67 |
| | | MrRoPE-Pro | **16.7** | **15.4** | **18.9** | **26.8** | **40.0** | **6.67** |

**(b)Qwen2.5-3B-Instruct**

| Context Window | Scale (S) | Extension Method | SD | LD | MD | LICL | LSD | CU |
|---|---|---|---|---|---|---|---|---|
| 32K | 4 | YaRN | 15.8 | 10.2 | 17.6 | 25 | 50.0 | 6.67 |
| | | MrRoPE-Pro | **16.5** | **12.8** | **17.6** | **27.5** | **50.0** | **6.67** |

writing, as well as the standardization of punctuation to enhance the manuscript's academic rigor and authority."

# E LIMITATION

Our RoPE-extension method primarily focuses on a training-free extension of the context window. While effective, the absence of fine-tuning experiments limits direct comparisons with other extension methods like xPOS and LongRoPE (Sun et al., 2022; Hua et al., 2024), which are often evaluated in a fine-tuning setting. Incorporating such experiments in the future would more comprehensively demonstrate the superiority and integration potential of our method.

Furthermore, the current theoretical framework and its advantages are intrinsically tied to the RoPE mechanism. The generalizability of the core "radix transformation" concept to other positional encoding schemes (e.g., T5's relative bias, ALibi) (Raffel et al., 2020; Press et al., 2021) remains an open question, which may limit the broader applicability of the approach beyond transformer architectures employing RoPE.

