# OpenReview forum: "MrRoPE: Mixed-radix Rotary Position Embedding"
_ICLR.cc/2026/Conference — ICLR 2026 Oral_

### Official Review · Reviewer_papb · 2025-10-29

**Soundness:** 3
**Presentation:** 4
**Contribution:** 3
**Rating:** 8
**Confidence:** 4

**Summary:**

This paper introduces MrRoPE (Mixed-Radix Rotary Position Embedding), a training-free method for extending the context length of transformer models equipped with RoPE (Rotary Position Embeddings). The key insight is to reinterpret RoPE as a form of mixed-radix number system, where each rotational frequency corresponds to a different positional base. The authors propose two practical schemes: 1) MrRoPE-Uni: uniformly redistributes intermediate rotation frequencies; 2)MrRoPE-Pro: progressively adjusts them across dimensions. The authors show that these schemes unify and generalize several prior test-time scaling methods such as NTK and YaRN. Extensive experiments on LLaMA-2/3 and Qwen models demonstrate consistent improvements across multiple long-context benchmarks (RULER, InfiniteBench, LongBench-v2, Needle-in-a-Haystack) without any fine-tuning. MrRoPE-Pro, in particular, exhibits stronger stability and better RoPE-bound behavior at very long sequence lengths (up to 128K tokens).

**Strengths:**

1. **Novel theoretical perspective**: The mixed-radix interpretation provides a new and elegant way to understand RoPE scaling. It explains existing methods (NTK, YaRN) under a unified mathematical framework and motivates new variants.
2. **Practical and lightweight**: The proposed method is training-free and only modifies the RoPE computation at inference time. It can be easily applied to off-the-shelf models, offering immediate practical value.
3. **Comprehensive empirical evaluation**: Experiments are extensive, covering perplexity, synthetic retrieval, and multiple real-world long-context benchmarks. The improvements of MrRoPE-Pro over prior inference-only methods are consistent and significant.
4. **Interpretability and diagnostics**: The paper connects the improvements to theoretical properties such as the “RoPE bound” and attention distribution behavior, providing an interpretable explanation of why progressive scaling helps.
5. **Strong empirical gains without fine-tuning**: MrRoPE-Pro achieves competitive or superior performance compared to other test-time extensions and even approaches the performance of some fine-tuned long-context models.

**Weaknesses:**

1. **Lack of strictly rigorous theoretical analysis**: The “RoPE as radix conversion” formulation relies on approximations (ignoring floor/mod operations), but the paper does not formally analyze the approximation error or its effect on attention stability. The derivations remain mostly intuitive rather than strictly formal proven.
2. **Missing fine-tuned or stronger baselines**: The comparison is limited to test-time methods (YaRN, NTK). Including light fine-tuning baselines (e.g., LongRoPE, xPOS, or other retrained RoPE variants) would help position the method more clearly in terms of cost–performance trade-off.
3. **Hyperparameter sensitivity not analyzed**: MrRoPE-Pro depends on specific tuning of hyper-parameters. The paper does not present an ablation or sensitivity study, leaving unclear whether performance is robust across models or sequence lengths.
4. **No evaluation of runtime or stability**: Since MrRoPE modifies rotation frequency computation, the paper should report inference-time latency, GPU overhead, and numerical stability (especially under FP16 inference, quantized inference is also welcomed).

**Questions:**

1. How accurate is the approximation of RoPE as a mixed-radix system when floor/mod terms are **non-negligible**? Could you provide error bounds or an empirical validation of this assumption?
2. How sensitive is MrRoPE-Pro to the choice of scaling parameters and dimensional splits? Would the same configuration generalize to very different models or hidden sizes?
3. Could unified radix conversion be extended beyond RoPE? For instance, to ALiBi or other relative positional encodings? Is the mixed-radix perspective specific to sinusoidal embeddings?
4. What is the inference-time cost or memory overhead compared to baseline RoPE and YaRN implementations?
5. Could you provide results for fine-tuned long-context methods to better contextualize MrRoPE’s efficiency vs. performance trade-off?

---

> ### Author Response · Authors · 2025-11-21
> **Response to Reviewer papb [1/2]**
>
> Thank you for your thoughtful and thorough review of our work.
>
> **[W1&Q1. ] "Regarding the floor/mod operation"**
>
> > Thank you for for raising this insightful question. Like we claimed in **sec 2.2**, when we keep the floor&mod operation, the outcome of rotation angle is the real position $n$. Therefore, we can first define an error $e$ = $\hat{S}(n)- n$，where $\hat{S}(n) = \sum^D_{d=1}((\frac{n}{\beta^{d-1}} \space \text{mod} \space 2\pi)·\beta^{d-1})$ is a biased position based on RoPE. Apparently, our goal is find a bound of $e$. For each digit $d$, define
> > $$
> > a_d=\left(\frac{n}{\beta^{d-1}} \bmod 2\pi\right)\in[0,2\pi).
> > $$
> > Then there exists an integer $k_d$ such that
> >
> > $$
> > \frac{n}{\beta^{d-1}} = a_d + 2\pi k_d .
> > $$
> > Hence the $d$-th term of the biased estimator satisfies
> >
> > $$
> > a_d\beta^{d-1} = n - 2\pi k_d\beta^{d-1}.
> > $$
> > Summing over all $d=1,\ldots,D$,
> >
> > $$
> > \hat S(n)
> > = \sum_{d=1}^D \left(n - 2\pi k_d\beta^{d-1}\right)
> > = Dn - 2\pi\sum_{d=1}^D k_d\beta^{d-1}.
> > $$
> > writing  $q_d = \frac{n}{2\pi \beta^{d-1}}$ yields
> > $$
> > k_d\beta^{d-1}
> > = \frac{n}{2\pi} - \beta^{d-1}\left[q_d\right],
> > $$
> > where $\left[q_d\right]\in[0,1)$ denotes the fractional part. Substituting into the previous expression causes the constant terms to cancel, giving the compact form of e
> >
> > $$
> > e = \hat{S}(n) -n = 2\pi \sum_{d=2}^{D} \beta^{d-1} \left[\frac{n}{2\pi \beta^{d-1}}\right] .
> > $$
> > Considered that $0\leq \left[\frac{n}{2\pi\beta^{d-1}}\right]\leq\frac{n}{2\pi\beta^{d-1}}$ ，its easy to get a simple bound of e, where $0 \leq e \leq (D-1)n$. This bound explains why does the radix theory run well in the early positions, and exhibit a linear trend with the increasment of $\beta$, just like what we presented in paper's figure 2.
>
> **[W3&Q2. ] "Hyperparameters sensitivity"**
>
> > Thank you for for raising this point. We would like to clarify that **our method consistently outperforms YaRN under all hyperparameter pairs**, and to highlight that the gains come from the method itself rather than parameter changing, we directly followed the optimal configuration reported in the YaRN paper. This avoids introducing additional confounding variables and ensures a fully fair comparison.
> >
> > To further analysis the effect of different hyperparameter choices, we added a further ablation experiment in the **Revised PDF-Appendix B.1**, which demonstrated that:
> >
> > 1. Both YaRN and MrRoPE are not sensitive to the choice of $\alpha$ and $\beta$. When $\alpha$ or $\beta$ varies within a small range, the resulting changes in PPL are minimal.
> > 2. Models with different architectures/hidden-size may have different optimal hyperparameter choices, but $\alpha = 32$ and $\beta = 1$ are almost always near-optimal or optimal.
>
> **[Q3. ] "Could unified radix theory extended to other PE methods?"**
>
> > We thank the reviewer for this insightful question. The answer is yes. Our radix framework connects RoPE’s frequency adjustments to mixed-radix transformations, and thus can be applied to any RoPE-based method that rescales frequencies (e.g., NTK-aware interpolation, PI, YaRN, LongRoPE, ...).
> >
> > In contrast, schemes that do not alter rotary frequencies fall outside this scope. For instance,
> >
> > - ALiBi, which does not rely on RoPE at all and thus falls outside the scope of radix theory.
> > - xPos, while built upon RoPE, does not alter frequencies but instead applies scaling to the q/k vectors directly, thus falls outside the scope of radix theory.
> >
> > Notably, methods like LongRoPE/LongRoPE2, which learn frequency adjustments during fine-tuning, can also be viewed as implicitly searching for an optimized radix transformation strategy.
> >
> > In summary, **MrRoPE has the potential to be applied to any methods, who is 1)based on RoPE , 2)try to find a better frequency combination.**
>
> **[Q4. ] "Inference time cost"**
>
> > Our method does not introduce any inference latency. It is a one-time deterministic pre-computation of rotation freqencies for each dimensions (Only happened with RotaryEmbedding's initialization); once initialized, the model’s forward pass and FLOPs are identical to standard RoPE. Thus, neither the inference-time cost or memory will increase compared to RoPE.

---

> ### Author Response · Authors · 2025-11-21
> **Response to Reviewer papb [2/2]**
>
> **[W4&Q5. ] "efficiency v.s. performance"**
>
> > We thank the reviewer for raising this point. For a concrete comparison, we fine-tuned LLaMA2-7B on PG19 using 8×A100 GPUs to reach an 8K context window. After 800 steps, the model achieved a PPL of 3.47, whereas our training-free MrRoPE-Pro reached 4.66.
> >
> > Although fine-tuning yields lower perplexity, it required 11 hours on 8xA100 just to reach 8K. Extending modern 30B/70B/100B-scale models to 128K contexts would be computationally prohibitive.
> >
> > This cost-performance gap underscores the value of our method: MrRoPE-Pro delivers meaningful long-context gains with zero training cost, making it a practical and economical alternative in real-world deployments.
>
> **[W2. ] "Lacks of fine-tuning experiments"**
>
> > Thank you for for raising this point. We omitted fine-tuning experiments from our paper for two primary reasons:
> >
> > 1. First, as modern LLMs already provide large native context windows (e.g., 32K), further fine-tuning for length extension is computationally expensive, making a training-free approach more practical and impactful.
> > 2. Second, although fine-tuning can offer additional comparisons, it also introduces data bias and optimization noise that obscure the behavior of the positional encoding itself. To cleanly validate our radix theory and isolate the inherently superior scaling strategy, we therefore adopt a training-free setting.
> >
> > To validate the potential of MrRoPE-Pro in fine-tuning scenario, we trained a Llama2-7b-hf model into 8K context window (Due to the computational limitations, that's extreme length we can do) , using both MrRoPE-Pro and YaRN respectively. Follows is the result of perplexity scores using a sliding-window(8K) across various context length.
> >
> > | Extension Method | Context Window | 32K      | 64K      | 96K      | 128K     |
> > | ---------------- | -------------- | -------- | -------- | -------- | -------- |
> > | YaRN(s=2)        | 8K             | 2.96     | 3.15     | 3.32     | 3.68     |
> > | MrRoPE-Pro(s=2)  | 8K             | **2.87** | **3.02** | **3.19** | **3.47** |
> >
> > Apparently, our method also demonstrated a great potential in fine-tuning scenerios.
>
> **We sincerely look forward to engaging in further discussions.**

---

> ### Comment · Reviewer_papb · 2025-11-24
>
> Thanks for your author responses. This is a good paper with both solid analytical derivations and detailed experiments. My questions have been solved.
>
> Looking forward to using MrRoPE in the near future (Maybe integrated into open-source repos like *Huggingface Transformers* after your paper's acceptance).

---

### Official Review · Reviewer_on7t · 2025-10-30

**Soundness:** 3
**Presentation:** 3
**Contribution:** 2
**Rating:** 6
**Confidence:** 4

**Summary:**

This paper reframes the problem of extending the context window of LLMs with RoPE, to view position encoding through the lens of mixed radix conversion. This allows to unify previous methods as different choices within the proposed framework and outperform YaRN on 13 tasks.

**Strengths:**

The unifying theory is good and the results are compelling across different tasks. Good to see a training free method has consistent gains.

**Weaknesses:**

It is not clear if this radix idea can work for other positional encoding method, which limits the impact to RoPE.  Also not clear if the progressive scaling in the proposed method introduces any latency compared to YaRN?

Would also be helpful to have sensitive analysis, about the parameters inherited from YaRN in appendix B.

**Questions:**

Would you explain why the progressive strategy in the proposed method is better than YaRN's way? Is there any inference latency penalty for using the proposed method or YaRN? If the radix concept could be used for other types of positional encodings?

---

> ### Author Response · Authors · 2025-11-21
> **Response to reviewer on7t**
>
> Thank you for your thoughtful and thorough review of our work.
>
> **[Q1. ] "Explain why the progressive strategy is better than YaRN"**
>
> > The progressive strategy outperforms YaRN because it preserves the high-frequency components in mid-range dimensions that YaRN’s regressive scaling suppresses. By allocating stronger scaling to later dimensions, it achieves a better balance between high- and low-frequency information, leading to more stable and effective extrapolation.
>
> **[W2&Q2. ] "Is there any inference latency penalty? "**
>
> > Our method does not introduce any inference latency. It is a one-time deterministic pre-computation of rotation freqencies for each dimensions (Only happened with RotaryEmbedding's initialization); once initialized, the model’s forward pass and FLOPs are identical to standard RoPE. Thus, inference speed remains unchanged.
>
> **[W1&Q3. ] "If the radix concept can be used for other PE methods? "**
>
> > We thank the reviewer for this insightful question. We would like to clarify that our radix framework connects RoPE’s frequency adjustments to mixed-radix transformations, and thus can be applied to any RoPE-based method that rescales frequencies (e.g., NTK-aware interpolation, PI, YaRN...).
> >
> > In contrast, schemes that do not alter rotary frequencies fall outside this scope. For instance,
> >
> > - ALiBi, which does not rely on RoPE at all and thus falls outside the scope of radix theory.
> > - xPos, while built upon RoPE, does not alter frequencies but instead applies scaling to the q/k vectors directly, thus falls outside the scope of radix theory.
> >
> > Notably, methods like LongRoPE/LongRoPE2, which learn frequency adjustments during fine-tuning, can be viewed as implicitly searching for an optimized radix transformation strategy.
> >
> > In summary, MrRoPE has the potential to be applied to any methods, who is 1)based on RoPE , 2)try to find a better frequency combination.
>
> **[W3. ] "Lacks a sensitivity analysis of hyperparameters $\alpha$ and $\beta$ "**
>
> > Thank you for for raising this point. We would like to clarify that **our method consistently outperforms YaRN under all hyperparameter pairs**, and to highlight that the gains come from the method itself rather than parameter changing, we directly followed the optimal configuration reported in the YaRN paper. This avoids introducing additional confounding variables and ensures a fully fair comparison.
> >
> > To further analysis the effect of different hyperparameter choices, we added a further ablation experiment in the **Revised PDF-Appendix B.1**, which demonstrated that:
> >
> > 1. Both YaRN and MrRoPE are not sensitive to the choice of $\alpha$ and $\beta$. When $\alpha$ or $\beta$ varies within a small range, the resulting changes in PPL are minimal.
> > 2. Models with different architectures may have different optimal hyperparameter choices, but $\alpha = 32$ and $\beta = 1$ are almost always near-optimal or optimal.
>
> **We sincerely look forward to engaging in further discussions.**

---

> ### Author Response · Authors · 2025-11-27
>
> Dear Reviewer on7t,
>
> We hope this message finds you well. We sincerely thank you for your meticulous and insightful feedback on our manuscript. Your valuable comments have significantly improved the quality of our work, and we greatly appreciate the time and dedication you devoted to reviewing our paper.
>
> However, as the discussion period is nearing its end and we have posted our reply for a few days, we wanted to ensure we have addressed all your concerns satisfactorily. If there are any additional points or feedback you'd like us to consider please let us know.Your insights are invaluable to us, and we're eager to address any remaining issues to improve our work.
>
> Thank you for your time and effort in reviewing our paper.

---

### Official Review · Reviewer_1qVM · 2025-11-02

**Soundness:** 3
**Presentation:** 2
**Contribution:** 2
**Rating:** 6
**Confidence:** 2

**Summary:**

This paper focuses on the context window extension problem of rotational position encoding (RoPE). Aiming at the pain point that the existing RoPE extension strategies are diverse and lack a unified theoretical foundation, the paper proposes a mixed-base rotational position encoding (MrRoPE) framework, and develops an efficient training-independent extension method based on it, and the related results are verified by quantitative experiments.

**Strengths:**

- MrRoPE constructs a universal framework from the radix conversion perspective, and unifies mainstream RoPE extension methods such as NTK and YaRN into different base conversion strategies, which solves the problem of the lack of a unified theoretical basis for the existing extension schemes, and facilitates system analysis and optimization.
- The proposed MrRoPE-Pro requires no additional fine-tuning and performs outstandingly in long context tasks: over 85% recall in Needle-in-a-Haystack tests with 128K contexts, more than twice the accuracy of YaRN in Infinite-Bench retrieval and dialog tasks, and in benchmarks such as RULER, LongBench-V2, and others. and consistently outperforms existing methods in benchmarks such as RULER, LongBench-V2, etc.
- The upper bound on the effective coding length of RoPE is theoretically boosted, and the model maintains excellent performance over the full context range of 8K to 128K through a progressive base conversion strategy that preserves the fine-grained information in the high-frequency dimensions and stabilizes the distribution of the attention scores in the intermediate dimensions.

**Weaknesses:**

- The RoPE extension method (MrRoPE-Uni/Pro) proposed in this paper only focuses on the “no-training” context-window extension mode, and no fine-tuning experiments have been conducted. This makes it impossible to make a direct comparison with extension methods such as xPOS and LongRoPE, which need to be evaluated in fine-tuning scenarios, and it is difficult to comprehensively prove its superiority in different application scenarios and its integration potential with other methods.
- It is well known that the training process of LLM model is complex and cumbersome, and the variation of hyperparameters may substantially affect the training results, so the authors should explore in detail whether the choice of hyperparameters affects the model performance, and whether the same hyperparameters proposed in the paper can be used for LLMs with different architectures.

**Questions:**

see weakness

---

> ### Author Response · Authors · 2025-11-21
> **Response to reviewer 1qVM**
>
> Thank you for your thoughtful and thorough review of our work.
>
> **[W1. ] "Lacks of fine-tuning experiments"**
>
> > Thank you for for raising this point. We omitted fine-tuning experiments from our paper for two primary reasons:
> >
> > 1. First, as modern LLMs already provide large native context windows (e.g., 32K), further fine-tuning for length extension is computationally expensive, making a training-free approach more practical and impactful.
> > 2. Second, although fine-tuning can offer additional comparisons, it also introduces data bias and optimization noise that obscure the behavior of the positional encoding itself. To cleanly validate our radix theory and isolate the inherently superior scaling strategy, we therefore adopt a training-free setting.
> >
> > To validate the potential of MrRoPE-Pro in fine-tuning scenario, we trained a Llama2-7b-hf model into 8K context window (Due to the computational limitations, that's extreme length we can do) , using both MrRoPE-Pro and YaRN respectively. Follows is the result of perplexity scores using a sliding-window(8K) across various context length.
> >
> > | Extension Method | Context Window | 32K      | 64K      | 96K      | 128K     |
> > | ---------------- | -------------- | -------- | -------- | -------- | -------- |
> > | YaRN(s=2)        | 8K             | 2.96     | 3.15     | 3.32     | 3.68     |
> > | MrRoPE-Pro(s=2)  | 8K             | **2.87** | **3.02** | **3.19** | **3.47** |
> >
> > Apparently, our method also demonstrated a great potential in fine-tuning scenerios.
>
> **[W2. ] "Hyperparameter experiments"**
>
> > Thank you for for raising this point. We would like to clarify that **our method consistently outperforms YaRN under all hyperparameter pairs**, and to highlight that the gains come from the method itself rather than parameter changing, we directly followed the optimal configuration reported in the YaRN paper. This avoids introducing additional confounding variables and ensures a fully fair comparison.
> >
> > To further analysis the effect of different hyperparameter choices, we added an ablation experiment in the **Revised PDF-Appendix B.1**, which demonstrated that:
> >
> > 1. Both YaRN and MrRoPE are not sensitive to the choice of $\alpha$ and $\beta$. When $\alpha$ or $\beta$ varies within a small range, the resulting changes in PPL are minimal.
> > 2. Models with different architectures may have different optimal hyperparameter choices, but $\alpha = 32$ and $\beta = 1$ are almost always near-optimal or optimal.
>
> **We sincerely look forward to engaging in further discussions.**

---

> ### Author Response · Authors · 2025-11-27
>
> Dear Reviewer 1qVM,
>
> We hope this message finds you well. We sincerely thank you for your meticulous and insightful feedback on our manuscript. Your valuable comments have significantly improved the quality of our work, and we greatly appreciate the time and dedication you devoted to reviewing our paper.
>
> However, as the discussion period is nearing its end and we have posted our reply for a few days, we wanted to ensure we have addressed all your concerns satisfactorily. If there are any additional points or feedback you'd like us to consider please let us know.Your insights are invaluable to us, and we're eager to address any remaining issues to improve our work.
>
> Thank you for your time and effort in reviewing our paper.

---

> > ### Comment · Reviewer_1qVM · 2025-11-28
> >
> > Thank you for the additional information which has explained my doubts, I am appreciative of the work done by all of you and I will keep my rating the same.

---

### Official Review · Reviewer_oZP9 · 2025-11-03

**Soundness:** 2
**Presentation:** 3
**Contribution:** 2
**Rating:** 6
**Confidence:** 3

**Summary:**

This paper aims to provide a unified theoretical foundation for context window extension of Rotary Position Embedding (RoPE) in LLMs. It proposes the MrRoPE (Mixed-radix RoPE) framework, which reinterprets RoPE and its extensions as a mixed-radix number system conversion. Based on this, the paper introduces a new training-free method, MrRoPE-Pro (progressive radix conversion), and demonstrates its significant advantages over state-of-the-art methods like YaRN across multiple benchmarks.

**Strengths:**

The work addresses the critical problem of training-free context extension for LLMs, and its unifying framework could have a profound impact on future research. The connection between RoPE extension and radix conversion theory is a highly novel and insightful perspective, elevating heuristic designs to a theoretical level.

**Weaknesses:**

1. The characterization of YaRN as "regressive" is based on observation rather than rigorous derivation. The paper fails to mathematically prove this from YaRN's original equations.
2. The method heavily relies on hyperparameters (`α`, `β`) inherited from YaRN, yet lacks a sensitivity analysis or ablation study to verify their optimality for the new method.
3. The comparison is limited to training-free RoPE methods, failing to adequately position the work relative to fine-tuning-based approaches or alternative schemes like ALiBi.
4. Experiments are confined to small- to medium-scale models (3B-8B). The generalizability of its conclusions to larger models (e.g., 70B scale) remains unproven.
5. The analysis of downstream performance gains is superficial. It reports *what* improved but lacks qualitative or error analysis to explain *why* MrRoPE-Pro is better.
6. The method's potential as a better initialization for long-context fine-tuning is untested. A crucial experiment is missing to verify if its advantage persists after fine-tuning.

**Questions:**

1.  Could you provide a more rigorous mathematical derivation, starting from YaRN's original interpolation formulas, to derive its corresponding scaling factors `λj`?
2.  Have you conducted any sensitivity analysis or ablation studies on the hyperparameters `α` and `β` inherited from YaRN?
3.  Could you elaborate on the fundamental trade-offs among the regressive (YaRN), uniform (Uni), and progressive (Pro) strategies?
4.  Does your "progressive" strategy inherently maintain attention score stability better than YaRN, reducing the reliance on heuristics like temperature scaling `t`?

---

> ### Author Response · Authors · 2025-11-21
> **Response to reviewer oZP9 [1/2]**
>
> Thank you for your thoughtful and thorough review of our work.
>
> **[W1&Q1. ] "YaRN is treated as regressive-conversion based on observation rather than rigorous derivation."**
>
> > Thank you for your feedback. We apologize if the presentation in our paper was not sufficiently clear. In fact, YaRN’s categorization can be established rigorously and elegantly. Due to space limitations, we provide a complete derivation about that in **Revised PDF-Appendix A.2.1**. Briefly, we show that: for all rope-theta settings where $b>1$ or $0<b<1$, YaRN's scale factors always satisfy $\lambda_{j−1}>\lambda_{j}$ , which characterizes YaRN as a regressive conversion. Notably, the only degenerate case arises when base $b=1$, in which YaRN collapses to the uniform class; however, this setting is not applicable to RoPE situation, and therefore YaRN can be definitively categorized as regressive.
> >
> > Thank you for pointing this out, it helps us to revise our paper.
>
> **[W2&Q2. ] "Lacks a sensitivity analysis of hyperparameters $\alpha$ and $\beta$ "**
>
> > Thank you for for raising this point. We would like to clarify that **our method consistently outperforms YaRN under all hyperparameter pairs**, and to highlight that the gains come from the method itself rather than parameter changing, we directly followed the optimal configuration reported in the YaRN paper. This avoids introducing additional confounding variables and ensures a fully fair comparison.
> >
> > To further analysis the effect of different hyperparameter choices, we added an ablation experiment in the **Revised PDF-Appendix B.1**, which demonstrated that:
> >
> > 1. Both YaRN and MrRoPE are not sensitive to the choice of $\alpha$ and $\beta$. When $\alpha$ or $\beta$ varies within a small range, the resulting changes in PPL are minimal.
> > 2. Models with different architectures may have different optimal hyperparameter choices, but $\alpha = 32$ and $\beta = 1$ are almost always near-optimal or optimal.
>
> **[W5&Q3. ] "Trade-offs between three kinds of conversion"**
>
> > We thank the reviewer for this insightful question. The essential trade-off lies in how each strategy treats the mid-range dimensions, which inherently mix both higher- and lower-frequency information. Under a fixed scaling sum S, emphasizing one type of information inevitably compromises the other. To address this issue,
> >
> > - **Regressive (YaRN)** applies strong radix scaling to early dimensions, pushing mid-range dimensions toward lower frequencies. This effectively preserves the low-frequency patterns learned during pre-training but reduces the retention of high-frequency details.
> > - **Progressive (Pro)** applies stronger scaling to later dimensions, prioritizing high-frequency preservation in the mid-range. However, this makes the lower-frequency components in these dimensions more prone to out-of-distribution behavior.
> > - **Uniform(Uni)** applies a same scaling factor to all dimensions, it's more like a simple baseline, which inherits both the advantages and disadvantages of Pro & Reg methods.
> >
> > In summary, the choice represents a trade-off between **in-distribution stability (YaRN)** and **out-of-distribution generalization (Our Method)**.
>
> **[Q4. ] "Regarding attention score and temperature t"**
>
> > We thank the reviewer for this question. Yes, our progressive strategy inherently provides more stable attention scores than YaRN, particularly in mid-range dimensions, as what we have shown in **Sec 4.4**. However, if we omit the temperature scaling factor $t$, an interesting phenomenon emerges: on older models like LLaMA2, the PPL of both YaRN and MrRoPE-Pro becomes almost unusable (PPL > 50 when scaling to 16× length), whereas on newer models like LLaMA3-8B, their performance actually improves (i.e., PPL of MrRoPE-Pro: 2.34 $\rightarrow$ 2.16, PPL of YaRN: 2.38 $\rightarrow$ 2.20). Based on this observation and to ensure the generality of the method, we conclude that the scaling factor $t$ remains irreplaceable for now. As for the reason of this interesting phenomenon, we need time to analyze it in the future.

---

> ### Author Response · Authors · 2025-11-21
> **Response to reviewer oZP9 [2/2]**
>
> **[W3&W4&W6. ] "Fine-tuning and compare with other models."**
>
> > Thank you for your valuable feedback on our paper. **Regarding W4**, due to limited computational resources, we are currently unable to run long-context evaluations on 70B models. **Regarding W6**, we would like to clarify that our method is designed to extend a RoPE-initialized model, so the pre-training scenario may not be suitable for evaluation. However, we are happy to provide a small fine-tuning experiment on a smaller model for your reference. We trained a Llama2-7b-hf model into 8K context window (Due to the computational limitations, that's extreme length we can do) , using both MrRoPE-Pro and YaRN respectively. Follow table shows the perplexity scores using a sliding-window(8K) across various evaluation context length.
> >
> > | Extension Method | Context Window | 32K      | 64K      | 96K      | 128K     |
> > | ---------------- | -------------- | -------- | -------- | -------- | -------- |
> > | YaRN(s=2)        | 8K             | 2.96     | 3.15     | 3.32     | 3.68     |
> > | MrRoPE-Pro(s=2)  | 8K             | **2.87** | **3.02** | **3.19** | **3.47** |
> >
> > Although this experiment does not fully meet the requirements of long-context training (>64K), it apparently still demonstrates the strong potential of our method in training scenarios, which may address the concerns you raised in **W3.**
> >
> > Thanks to your feedback and we will do more fine-tuning experiment after we got enough computatinal resources.
>
> **We sincerely look forward to engaging in further discussions.**

---

> ### Author Response · Authors · 2025-11-27
>
> Dear Reviewer oZP9,
>
> We hope this message finds you well. We sincerely thank you for your meticulous and insightful feedback on our manuscript. Your valuable comments have significantly improved the quality of our work, and we greatly appreciate the time and dedication you devoted to reviewing our paper.
>
> However, as the discussion period is nearing its end and we have posted our reply for a few days, we wanted to ensure we have addressed all your concerns satisfactorily. If there are any additional points or feedback you'd like us to consider please let us know.Your insights are invaluable to us, and we're eager to address any remaining issues to improve our work.
>
> Thank you for your time and effort in reviewing our paper.

---

### Author Response · Authors · 2025-12-03
**Rebuttal Summary**

Dear ACs/SACs/PCs

We sincerely thank you for your time and effort in managing the review process for our paper. We provide a summary of the current status of our paper and the progress of our rebuttal.

---
## Strengths

All reviewers found our work is highly insightful and novel, which not only provide a profound unified theoretical framework but also a more efficient method for training-free context window extension. Reviewer `oZP9` described it as **"have a profound impact"**, while Reviewers `papb` described it as **"elegant"**.

In specific, reviewers highlighted several strengths:

- **MrRoPE theory provide a novel theoretical perspective for context extension.**

  (`mentioned by all reviewers`)

- **MrRoPE-Pro exhibits consistent superior results through an extensive and comprehensive empirical evaluation.**

  (`mentioned by all reviewers`)

- **Interpretability and diagnostics of MrRoPE's superiority are well analyzed.**

  (`mentioned by reviewer 1qVM, papb`)

---
## Weakness

All concerns raised during review were fully addressed during the rebuttal. However until Nov 28th, we only got the response from reviewer `1qVM` and `on7t`, who have maintained their original scores and expressed positive expectations for the future potential impact of this work.

In specific, reviewers's concern mainly focus on following questions:

- **Detailed derivation about YaRN's regressive features (oZP9).**

  We provide a complete derivation about that in `Revised PDF-Appendix A.2.1`, which shows that: for all rope-theta settings YaRN's scale factors always satisfy $\lambda_{j-1} > \lambda_{j} $.

- **Strictly rigorous theoretical analysis about floor&mod operation (papb).**

  We provide a detailed derivation about the distance between the original MrRoPE and that reintroduced operations, shows a consistent claim as our paper described.

- **Lack of ablation studies on the hyper parameters (oZP9, 1qVM, on7t, papb).**

  We added an ablation experiment in the `Revised PDF-Appendix B.1`, to show that our method consistently outperforms YaRN under all hyperparameter pairs.

- **Fintuning scenario evaluation (oZP9, 1qVM, papb).**

  Despite our work is designed for training-free scenario (we explained why its more crucial than tuning scenario in corresponding replies), we provide an extra fine-tuning experiment for reviewers, showing the strong potential of MrRoPE-Pro in training settings.

- **If the radix concept can be used for other PE methods (papb, on7t)**

  We explain about why MrRoPE has the potential to be applied to any methods, who is 1)based on RoPE , 2)try to find a better frequency combination.

---

## Revisions summary

- **[Content Highlighted in Blue]** Improved clarity of our work.
- **[Appendix A.2.1]** Strict derivation explains why is YaRN a regressive radix conversion.
- **[Appendix B.1]** A sensitive analysis of hyper-parameter choices.

---

## Final Remarks

All reviewer concerns were resolved during the rebuttal, and no new issues were raised afterward. We deeply appreciate the reviewers' and AC’s efforts and the opportunity to improve our work.



**Best regards**,

*Submission5551 Authors*

---

### Meta-Review · Area_Chair_e6FT · 2026-01-06

**Summary:**

The submission presents MrRoPE, a novel theoretical framework that interprets Rotary Position Embedding (RoPE) extension as a mixed-radix number system conversion. This perspective is highly valued by all reviewers for its ability to unify existing methods like NTK and YaRN under a single mathematical foundation. The proposed method, MrRoPE-Pro, demonstrates significant practical value as a training-free solution for context window extension.

While some concerns regarding the scale of the models tested (3B-8B) remain, the authors' computational constraints were noted, and the theoretical contributions and strong 8B results are considered sufficient for acceptance.

Argument for Oral:

This work introduces a novel theoretical framework that elegantly understands and unifies previous methods like NTK and YaRN (papb), boosts the upper bound on the effective coding length of RoPE (1qVM), achieves "outstanding" and "consistent" results without fine-tuning (1qVM), and offers "immediate practical value" to the community (papb). Hence, the paper offers the kind of conceptual clarity, theoretical grounding, and high-utility that the ICLR audience may value in an oral presentation.

**Reviewer Concerns:**

Reviewer oZP9:

- Characterizing YaRN as "regressive" was based on observation only: Addressed. The authors provided a formal derivation in the revised version.

- Lack of ablation on hyperparameters: Addressed. Authors added Appendix B.1, showing the method is robust across various settings.

- Trade-offs between three kinds of conversion: Addressed with clarifications.

- Generalizability to 70B+ models: Outstanding. Authors cited computational limitations.


Reviewer 1qVM:

All addressed according to reviewer comment.


Reviewer on7t:

- Unclear if it applies to non-RoPE methods: Addressed. Authors clarified it applies to any RoPE-based method seeking better frequency combinations (e.g., PI, NTK) but not to ALiBi.

- Concern that the method increases inference cost: Addressed. Authors confirmed rotation frequencies are pre-computed at initialization, resulting in zero latency penalty.

- Reviewer requested sensitivity analysis: Addressed.

Reviewer papb:

All addressed according to reviewer comment.

**Reviewer Scores:**

Reviewer oZP9: Likely maintain 6 as the scale concern remains.

Reviewer 1qVM: Remain 6 according to the reviewer.

Reviewer on7t: Likely remain 6 or increase to 8 as all concerns are addressed.

Reviewer papb: Remain 8 according to the reviewer.

---

### Decision · Program_Chairs · 2026-01-26

Accept (Oral)